

# Regional differences in an established population of invasive Indo-Pacific lionfish (*Pterois volitans* and *P. miles*) in south Florida

David R. Bryan[1,2], Jeremiah Blondeau[3], Ashley Siana[1] and Jerald S. Ault[1]

[1] Department of Marine Ecosystems and Society, Rosenstiel School of Marine and Atmospheric Science, University of Miami, Miami, FL, United States of America
[2] Pacific States Marine Fisheries Commission, Alaska Fisheries Science Center, National Marine Fisheries Service, National Oceanic and Atmospheric Administration, Seattle, WA, United States of America
[3] Southeast Fisheries Science Center, National Marine Fisheries Service, National Oceanic and Atmospheric Administration, Miami, FL, United States of America

Corresponding author
David R. Bryan,
david.bryan@noaa.gov

## ABSTRACT

About nine years ago (circa 2009), Indo-Pacific lionfishes (*Pterois volitans* and *P. miles*) invaded the south Florida coral reef ecosystem. During the intervening period of time, there has been substantial research on their biology, life history, demography, and habitat preferences; however, little is known concerning their regional population status and trends in the region. Here, we use a large-scale fisheries independent reef fish visual survey to investigate lionfish population status among three south Florida regions: Dry Tortugas, Florida Keys, and southeast Florida. Density estimates (ind ha$^{-1}$) have been relatively stable since 2012, and are lower than other areas reported in the western Atlantic and Caribbean Sea. Low, stable population densities in south Florida suggest there may be a natural mechanism for lionfish population control. In the Dry Tortugas, lionfish density in 2016 was significantly lower (0.6 ind ha$^{-1}$ $\pm$ 0.15 SE) than the two other south Florida regions. The Dry Tortugas region has the highest percentage of marine protected areas, the lowest level of exploitation, and thus the highest densities of potential lionfish predators and competitors. In the Florida Keys and southeast Florida in 2016, lionfish densities were greater (5.4 ind ha$^{-1}$ $\pm$ 1.0 SE and 9.0 $\pm$ 2.7 SE, respectively) than the Dry Tortugas. Fishing pressure on lionfish was higher in these two regions, but densities of several potential predators and competitors were substantially lower. Despite relatively low regional lionfish densities that can be attributed to some combination of fishing mortality and natural biocontrol, lionfish are still well established in the south Florida coral reef ecosystem, warranting continued concern.

## INTRODUCTION

Indo-Pacific lionfishes (*Pterois volitans* and *P. miles*) are the first non-native marine fish species to become established in the central western Atlantic (*Whitfield et al., 2002*; *Schofield, 2009*). Although there were scattered reports of lionfish in south Florida in

the late 1990s and 2000s, lionfish were not prominent until 2009, perhaps from a pulse of larval recruits from upstream sources in the Caribbean Sea (*Morris, 2009*; *Johnston & Purkis, 2011*; *Ruttenberg et al., 2012*). Shortly after the arrival of larval recruits, lionfish became established in south Florida through local reproduction coupled with continued larval recruitment from the eastern Gulf of Mexico, Meso-American, and Cuban reefs (*Schofield, 2009*; *Johnston & Purkis, 2015*). Following their arrival in south Florida, lionfish occurrence and relative abundance increased dramatically (*Ruttenberg et al., 2012*).

Broad environmental tolerances, relatively rapid growth and high survivorship of recruits have facilitated explosive population growth of lionfish throughout most of their introduced range (*Morris, 2009*; *Claydon, Calosso & Traiger, 2012*; *Edwards, Frazer & Jacoby, 2014*; *Gardner et al., 2015*). Lionfish are also highly effective predators capable of consuming large quantities of native fish and crustaceans (*Morris & Akins, 2009*; *Muñoz, Currin & Whitfield, 2011*; *Green et al., 2012*; *Côté et al., 2013*). Relative to similarly-sized native predators such as snappers and groupers, lionfish can contribute to substantial declines in prey abundance (*Albins & Hixon, 2008*; *Albins, 2012*). The direct effects of these declines are not limited to small-bodied reef fish species, as lionfish also eat the juveniles of larger-bodied mesopredators and grazers, including economically important species such as groupers, snappers, and grunts (*Morris & Akins, 2009*; *Côté et al., 2013*; *Dahl & Patterson, 2014*). Thus, there is significant concern that lionfish predation on native reef fishes may threaten coral reef ecosystems throughout its introduced range (*Albins & Hixon, 2008*; *Albins & Hixon, 2013*).

Most of the research on lionfish has been conducted at relatively small spatial scales, resulting in a dearth of information regarding lionfish population status on a larger regional scale. The southern Florida reef fish visual census (RVC) is an extensive, multi-agency probability-based survey, designed to estimate the size-structured abundance of more than 300 species of tropical coral reef fishes (*Brandt et al., 2009*; *Smith et al., 2011*). The RVC has been conducted throughout the south Florida coral reef ecosystem since 1979 and currently includes three regions: Dry Tortugas, Florida Keys, and southeast Florida (Fig. 1). Although lionfish appear to have successfully invaded and colonized south Florida (*Ruttenberg et al., 2012*), there has been scant information on temporal or spatial trends in lionfish population size since their establishment. In this study, we used the RVC data to test hypotheses concerning population trends in lionfish density since their arrival, and whether differences exist among the three south Florida regions. To better understand factors that may control lionfish population size, we investigated density estimates of potential lionfish predators and competitors, and regional differences in fishing intensities.

## MATERIALS AND METHODS

The RVC is a probability-based stratified random sampling survey used in a collaborative, multi-agency fisheries independent monitoring program that has been conducted since 1979 to obtain size-structured abundance estimates of reef fish populations in south Florida (*Brandt et al., 2009*; *Smith et al., 2011*). Beginning in 2014, the RVC was incorporated into the larger National Coral Reef Monitoring Program, and is now one facet of a broader coral

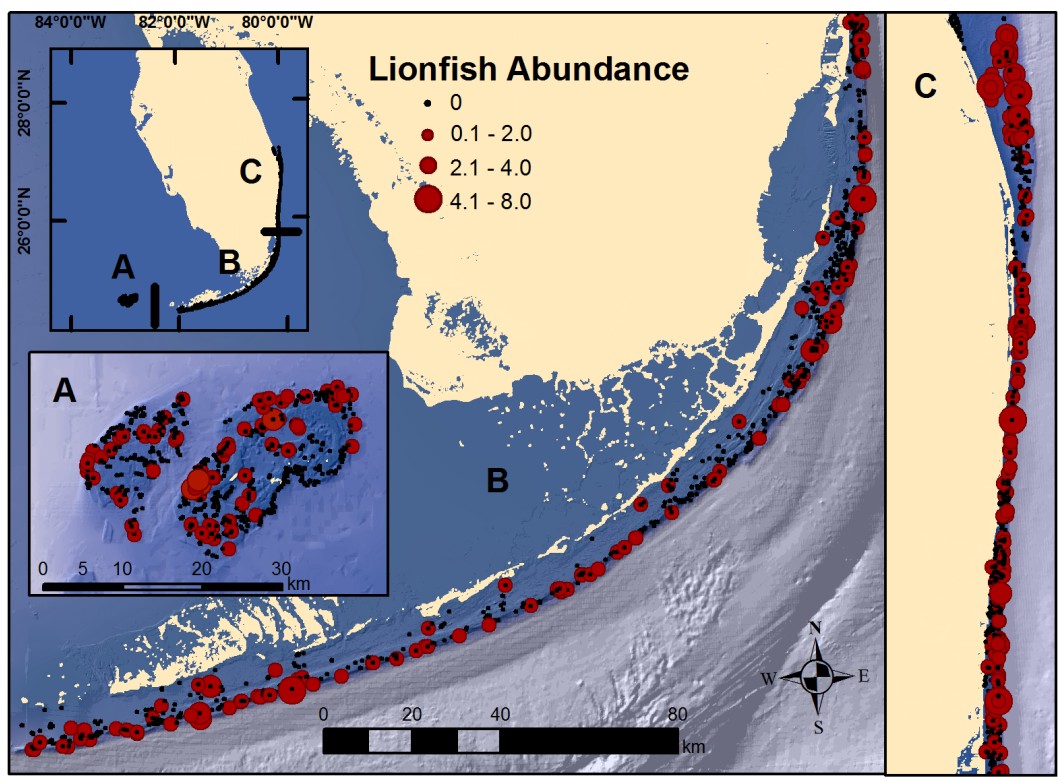

**Figure 1** **Map of south Florida regions.** (A) Dry Tortugas; (B) Florida Keys; and, (C) southeast Florida. Locations of RVC surveys and observed lionfish abundance at each secondary sampling unit during 2014 and 2016 surveys.

reef ecosystem monitoring effort that includes biological trends, climate-driven impacts and socioeconomic connections in the United States. Currently, the RVC is conducted bi-annually in three south Florida regions: (1) Dry Tortugas; (2) Florida Keys; and, (3) southeast Florida (Fig. 1). The spatial domain of the survey encompasses the full extent of mapped coral reef habitats to 35 m depths in each region.

The RVC uses a two-stage stratified random sampling design to partition the survey areas into subareas (i.e., strata) with varying levels of variance in reef fish density. Environmental features such as bathymetry and benthic habitat types were used to construct regional strata. A sampling frame, consisting of a finite number of non-overlapping primary sampling units (PSUs), was initially created by laying a 200 × 200 m grid over bathymetry and habitat maps. Recent updates in mapping products have allowed for a smaller 100 × 100 m grid to be used in southeast Florida, and in all regions in 2016. Each grid cell, was assigned to a strata based on the underlying habitat type and depth. The number and definition of strata vary between regions, as the Florida reef tract is characterized by a gradual shift in geology and habitat types (*Hoffmeister & Multer, 1968*; *Davis, 1982*; *Banks et al., 2007*).

The Dry Tortugas region is characterized by a large, relatively deep, western bank with extensive terrace reefs, occasional pinnacles, and ledges (*Franklin et al., 2003*). The Dry Tortugas region also contains the Dry Tortugas National Park, an atoll-like structure with

several shallow water banks, low to medium profile continuous reefs, low and high relief spur and groove formations, and both individual and aggregate patch reefs. In the Dry Tortugas, there are eight different strata comprised of three main habitat types with up to three levels of relief: isolated reefs with high, medium or low relief, continuous reefs with high, medium or low relief and spur and groove reefs with either high or low relief.

The Florida Keys region, which begins 100 km to the east of the Dry Tortugas, includes 250 km of platform reef characterized by gradual sloping forereef, low and high relief spur and groove, back reef and an extensive patch reef system. In the Florida Keys region, there are seven possible strata. The low relief forereef system is classified by three depth categories (<6 m, 6–18 m, and ≥18 m), the patch reef system is classified by location within the reef system (inshore, mid-channel, and offshore), and the last strata includes all high relief spur and groove habitats.

At the northern edge of the Florida Keys region, the classic tropical reef system shifts to a sponge and algae dominated reef constituted by several ridges that run parallel to the shoreline. In southeast Florida, there are eight habitat strata divided into low or high relief categories: a deep ridge strata in Martin County, linear reef along the outer reef tract including the deep ridge habitat outside of Martin County, shallow (<20 m) individual and aggregated patch reefs, deep (>20 m) individual and aggregated patch reefs, a deep ridge complex, linear reef along the middle reef tract, linear reef along the inner reef tract and the shallow ridge.

The size-structured abundance of reef fish were collected by trained scuba divers within a 15 m diameter cylinder (*Brandt et al., 2009*). A two-stage sampling scheme was employed to account for the disparity in area between a minimum mapping unit (40,000 m$^2$ to 10,000 m$^2$) of each PSU (grid cell) and the area surveyed by each diver (177 m$^2$), the second-stage unit (SSU). Within each PSU there were two SSUs. Because of diving safety concerns, each SSU was sampled by two closely spaced divers. For analysis, a single SSU sample was computed as the arithmetic average of the adjacent fish counts for paired divers.

Sample allocation among strata and site selection occured separately for each region. For each survey, a Neyman allocation scheme was used to determine the numbers of PSUs randomly selected for each stratum. This scheme accounts for the stratum's size and the standard deviation of density for several key species calculated from previous surveys. Thus, a stratum with a higher variance of fish density received a greater number of samples, as compared to its proportion of the total area in the region. Second-stage units (SSUs) were randomly selected in the field.

Estimation procedures for population density and variance from the two-stage stratified random sampling were adapted from *Cochran (1977)*, can be found in *Smith et al. (2011)*, and are as follows: First, the mean density for each PSU ($i$) in stratum ($h$) in region ($r$) was calculated by averaging the SSUs ($j$),

$$\overline{D}_{rhi} = \frac{1}{m_{rhi}} \sum_j D_{rhij}$$

where $m_{hi}$ is the number of SSUs sampled in PSU $i$ in stratum $h$. Regional stratum density was then calculated as the average of all PSUs in each stratum,

$$\overline{\overline{D}}_{rh} = \frac{1}{n_{rh}} \sum_i \overline{D}_{rhi}$$

where $n_{rh}$ is the number of PSUs sampled in stratum $h$ in region $r$. Finally the region-wide mean density estimate was calculated by summing the regional strata estimates that are weighted by area.

$$\overline{\overline{D}}_r = \sum_h w_{rh} \overline{\overline{D}}_{rh}.$$

The regional stratum weighting factor ($w_{rh}$) was generated by dividing the total number of possible SSUs in a stratum by the total number of SSUs in a region,

$$w_{rh} = \frac{N_{rh} M_{rh}}{\sum_h N_{rh} M_{rh}}$$

where $N_{rh}$ is the total possible number of PSUs in a stratum and $M_{rh}$ is the total possible number of SSUs per PSU in a stratum.

Estimation of variance for mean density began with calculating the sample variance among SSUs,

$$s_{2rh}^2 = \frac{1}{n_{rh}} \sum_i \left[ \frac{\sum_j \left( D_{rhij} - \overline{D}_{rhi} \right)^2}{m_{rhi} - 1} \right]$$

and the sample variance among PSUs.

$$s_{1rh}^2 = \frac{\sum_i \left( \overline{\overline{D}}_{rhi} - \overline{\overline{D}}_{rh} \right)^2}{n_{rh} - 1}.$$

Then the variance of mean density was calculated for each stratum as

$$\mathrm{var}\left[ \overline{\overline{D}}_{rh} \right] = \frac{\left( 1 - \frac{n_{rh}}{N_{rh}} \right)}{n_{rh}} s_{1rh}^2 + \frac{\frac{n_{rh}}{N_{rh}} \left( 1 - \frac{\overline{m}_{rh}}{M_{rh}} \right)}{n_{rh} m_{rh}} s_{2rh}^2$$

and finally, the variance of regional mean density was

$$\mathrm{var}\left[ \overline{\overline{D}}_r \right] = \sum_h w_{rh}^2 \mathrm{var}\left[ \overline{\overline{D}}_{rh} \right].$$

The standard error (SE) was calculated as the square root of the variance of the regional density estimate. The R package "RVC" was used to make the regional density and variance calculations (*Ganz, 2015*). In the RVC package, density is estimated at the SSU level (ind per 177 m$^2$), then converted to ind ha$^{-1}$ for consistency with most published lionfish research.

Annual estimates of mean regional lionfish population density were calculated for 2010, 2012, 2014, and 2016 for the Dry Tortugas, 2010, 2011, 2012, 2014, and 2016 for the Florida Keys, and 2012, 2014, 2015, 2016 for southeast Florida. Standard statistical

procedures were used to test for differences among years within a region and among regions for each year (*Lohr, 2010*). We constructed 95% confidence intervals (CI) for each estimate by multiplying the SE of each estimate by the appropriate values of the Student's *t*-distribution based on a 0.05 probability and the degrees of freedom. The degrees of freedom for a two-stage random sample design were calculated as the total number of SSUs minus the total number of PSUs and total number of strata. CI *t*-tests were used as they are better suited to sample design statistics and do not require homogenous variance of two distributions to test differences in mean responses. Trends in lionfish density were investigated within each region between years. Regional differences were compared for two regions at a time, by year. A significant difference in lionfish density ($p < 0.05$), between paired estimates, was defined as when the mean of one estimate did not fall into the confidence interval of the second estimate, and when the mean of the second estimate did not fall into confidence interval of the first estimate. Samples for the 2012 southeast Florida region were from 2012 and 2013. They were combined, since each year alone did not have enough samples to generate region-wide estimates. Data from 2014 and 2016 were combined, to increase the sample size and to calculate stratum level estimates of lionfish density for each region with a standard error. CI t-tests were used to compare stratum estimates within each region.

There are substantial differences in spatial protection of fisheries resources in the three regions. In the Dry Tortugas, 47.4% of the RVC survey domain is fully protected from fishing, while, in the Florida Keys only 4.4% of the survey domain is protected. There are no protected areas in the southeast Florida region. These differences, in combination with regional differences in geomorphology and benthic habitat types, can have an effect on the diversity and abundance of respective fish assemblages (*Beukers & Jones, 1998*; *Friedlander & Parrish, 1998*; *Gratwicke & Speight, 2005*). We evaluated the numerical importance of lionfish for each region in south Florida by calculating the individual densities for all reef fish species. A subset of piscivorous species that are potential predators and competitors was also evaluated (Table S1). The density of potential predators and piscivorous competitors was calculated for six groups for each region. These groups were based on taxa and fish size and included: snappers, jacks, large groupers, small groupers, hamlets, and *Serranus* spp. Differences in density of these groups among regions were compared with CI t-tests ($p < 0.05$).

## RESULTS

Since 2010, a total of 9,418 paired diver RVC surveys were conducted (Table 1). During this time, 1,663 lionfish were seen. There was no clear temporal trend in regional population density. Lionfish density in the Dry Tortugas was significantly greater in 2012 (2.2 ind ha$^{-1}$ ± 0.5 SE) and 2014 (2.6 ind ha$^{-1}$ ± 0.6 SE) than in 2010 (0.4 ind ha$^{-1}$ ± 0.3 SE), but in 2016 it declined to 0.6 ind ha$^{-1}$ (±0.15 SE) (Fig. 2). In the Florida Keys, density initially increased following the invasion to 6.5 ind ha$^{-1}$ (±1.2 SE) in 2011, declined significantly during 2012 and 2014, and then increased slightly to 5.3 ind ha$^{-1}$ (±1.0 SE) in 2016. Despite some fluctuations in density, there was no trend in the southeast Florida region

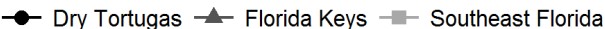

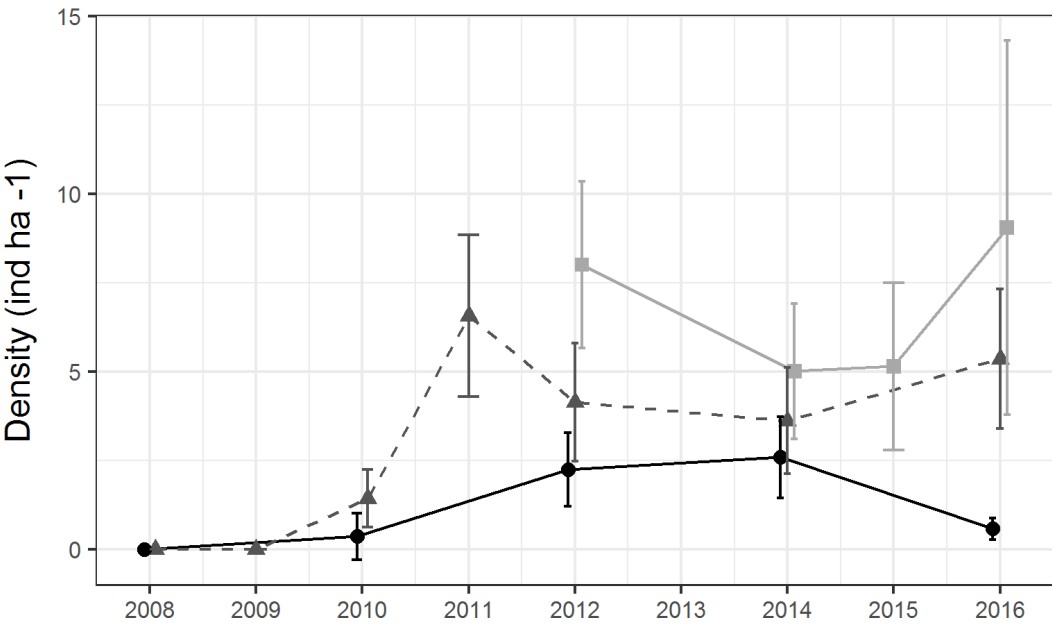

**Figure 2** **Lionfish population level density estimates (ind ha⁻¹) by region from 2008 through 2016.**
Bars show 95% confidence intervals.

**Table 1** **Number of secondary sample units (SSUs) surveyed and lionfish counted in each region since 2010.**

| | Dry Tortugas | | Florida Keys | | Southeast Florida | |
|---|---|---|---|---|---|---|
| Year | SSUs | Lionfish | SSUs | Lionfish | SSUs | Lionfish |
| 2010 | 703 | 4 | 740 | 37 | | |
| 2011 | | | 789 | 181 | | |
| 2012 | 813 | 136 | 803 | 91 | 1,073 | 390 |
| 2013 | | | | | | |
| 2014 | 704 | 111 | 860 | 88 | 605 | 156 |
| 2015 | | | | | 417 | 112 |
| 2016 | 544 | 55 | 797 | 121 | 570 | 181 |

since the survey began in 2012. Density in the southeast Florida region was significantly greater in 2012 (8.0 ind ha⁻¹ ± 1.2 SE) and in 2016 (9.0 ind ha⁻¹ ± 2.7 SE) than in 2014 and 2015 (5.0 ind ha⁻¹ ± 1.0 SE, 5.1 ind ha⁻¹ ± 1.2 SE, respectively). There were significant differences among regions, with lionfish density significantly lower in the Dry Tortugas than the southeast Florida region in 2012, 2014 and 2016. Dry Tortugas densities were also lower than the Florida Keys in 2012 and 2016. Lionfish density was significantly lower in the Florida Keys than southeast Florida in 2012, but there were no significant difference in 2014 and 2016 (Fig. 2).

There were differences in lionfish densities amongst strata within each region (Fig. 3). In the Dry Tortugas, isolated high relief strata had the highest density (11.1 ind ha$^{-1}$ $\pm$ 1.6 SE), and in general, higher relief strata had greater densities of lionfish. In the Florida Keys, the highest density was within the deep forereef strata (17.4 ind ha$^{-1}$ $\pm$ 3.8 SE) and densities in the other strata were similar. In southeast Florida, lionfish densities were the highest within the high and low relief deep ridge strata (51.5 ind ha$^{-1}$ $\pm$ 14.3 SE and 32.3 ind ha$^{-1}$ $\pm$ 23.4 SE, respectively), but were also high (>10 ind ha$^{-1}$) in other deep offshore strata. The strata with high densities of lionfish often comprised a minor proportion habitat in each region. For example, in the Dry Tortugas the isolated high relief strata with the highest density of lionfish made up only 2.5% of the hardbottom habitat in the survey domain (Fig. 3). In the Florida Keys, the deep forereef strata, which had the highest density of lionfish, made up 14.2% of the hardbottom habitat. In southeast Florida, the high and low relief deep ridge strata, which had the highest densities of lionfish, only represented 0.5% and 2.6% percent of hardbottom habitat in the survey, respectively.

In 2016, there were 133 species of fish in the Dry Tortugas with a greater abundance than lionfish out of total of 236 species observed. In the Florida Keys and southeast Florida, where 251 species were observed in 2016, lionfish were the 95th and 78th most abundant fish, respectively. Lionfish were the 30th most abundant piscivore in the Dry Tortugas out of 66 observed, 17th out of 66 in the Florida Keys, and 13th out of 68 in southeast Florida (Fig. 4). The densities of competitors and predators varied among regions, but in general, densities were higher in the Dry Tortugas. The Dry Tortugas had a significantly greater density of snappers (Lutjanidae), hamlets (*Hypoplectrus* spp.), and large groupers (Serranidae) than both the Florida Keys and southeast Florida (Fig. 5). There were significantly more *Serranus* spp. than the Florida Keys and small groupers than southeast Florida. The Florida Keys had a significantly greater density of small groupers than the other two regions, and more snappers and hamlets than southeast Florida. There was no differences in the densities of jacks among regions. Combined, there were 1041.0, 896.8, and 450.7 ind ha$^{-1}$ of competitors and predators from these groups in the Dry Tortugas, Florida Keys and southeast Florida regions, respectively.

## DISCUSSION

Following the initial increase in density after their 2009 invasion, lionfish populations in south Florida appear to have reached a relatively stable plateau within the coral reef fish community. Regional lionfish density estimates were generally lower in south Florida than those reported for other invaded areas using similar visual survey techniques: Belize, 160 ind ha$^{-1}$ (*Hackerott et al., 2013*); Cuba, 150 ind ha$^{-1}$ (*Hackerott et al., 2013*); Little Cayman Island, 21.5–162.5 ind ha$^{-1}$ (*Bejarano et al., 2014*); North Carolina, 84.6 ind ha$^{-1}$ (*Whitfield et al., 2014*); Panama, 300 ind ha$^{-1}$ (*Palmer et al., 2016*); and Venezuela, 30–121 ind ha$^{-1}$ (*Elise et al., 2015*). The south Florida estimates were similar to densities in their native range: Indian Ocean, 3.6 ind ha$^{-1}$ and Pacific Ocean, 0.27 ind ha$^{-1}$ (*Kulbicki et al., 2012*). This combination of low and stable densities suggests that there may be some combination of factors negatively influencing lionfish populations in south Florida. The factors controlling

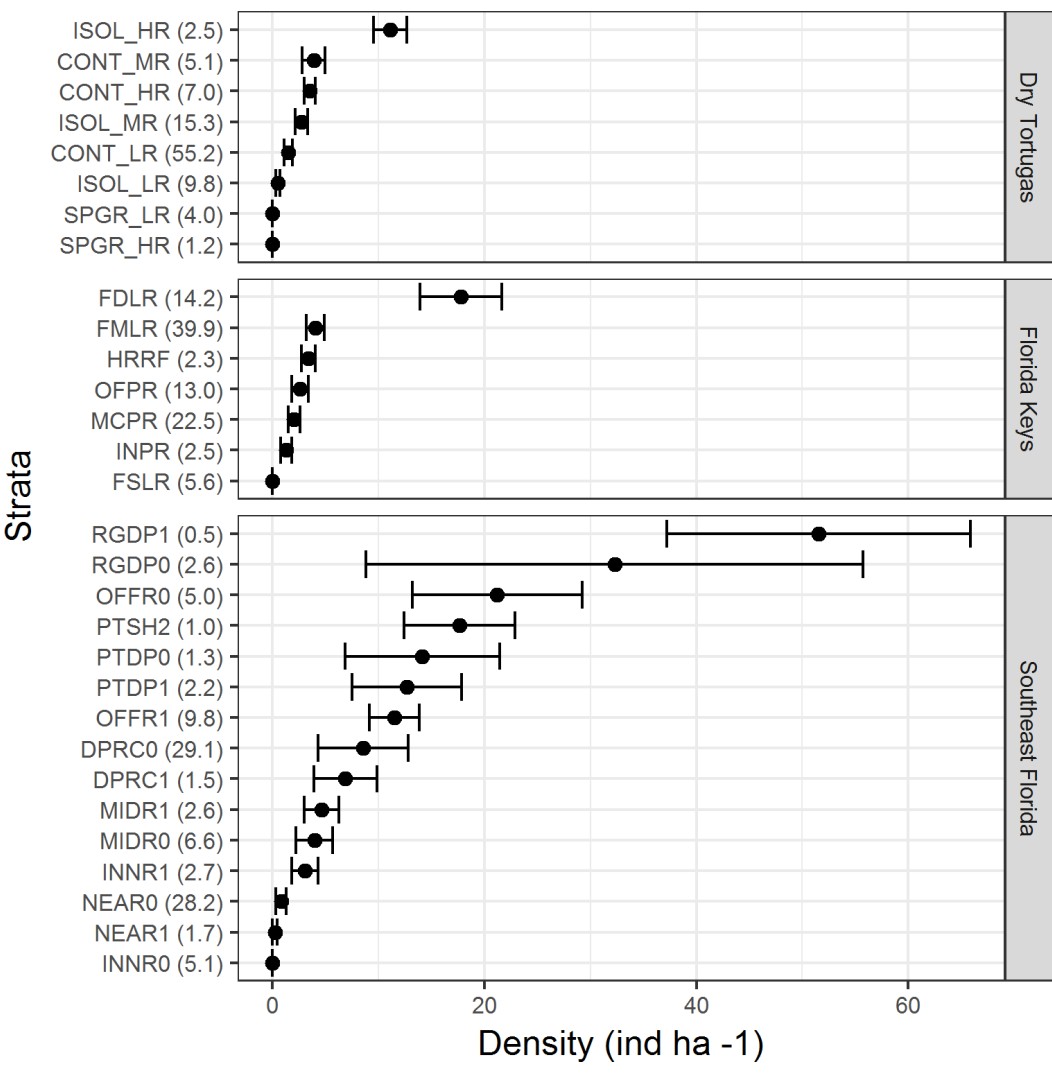

**Figure 3** **Lionfish density (ind ha$^{-1}$) by stratum for each region in 2014 and 2016 combined.** Bars represent standard error. Values in parentheses are the percentage of area within each region for that stratum. Strata abbreviations for the Dry Tortugas are: ISOL_HR (isolated high relief), CONT_MR (continuous medium relief), CONT_HR (continuous high relief), ISOL_MR (isolated medium relief), CONT_LR continuous low relief), ISOL_LR (isolated low relief), SPGR_LR (spur and groove low relief), SPGR_HR (spur and groove high relief); for the Florida Keys: FDLR (forereef ≥ 18 m depth), FMLR (forereef 6–18 m depth), HRRF (high relief spur and groove), OFPR (offshore patch reef), MCPR (mid-channel patch reef), INPR (inshore patch reef), and FSLR (forereef <6 m depth); for southeast Florida: RGDP1 (high relief deep ridge in Martin County), RGDP0 (low relief deep ridge in Martin County), OFFR0 (low relief outer linear reef), PTSH2 (shallow patch reef), PTDP0 (low relief deep patch reef), PTDP1 (high relief deep patch reef), OFFR1(high relief outer linear reef), DPRC0 (low relief deep ridge complex), DPRC1(high relief deep ridge complex), MIDR1 (high relief middle linear reef), MIDR0 (low relief middle linear reef), INNR1 (high relief inner linear reef), NEAR0 (low relief shallow ridge), NEAR1 (high relief shallow ridge), and INNR0 (low relief inner linear reef).

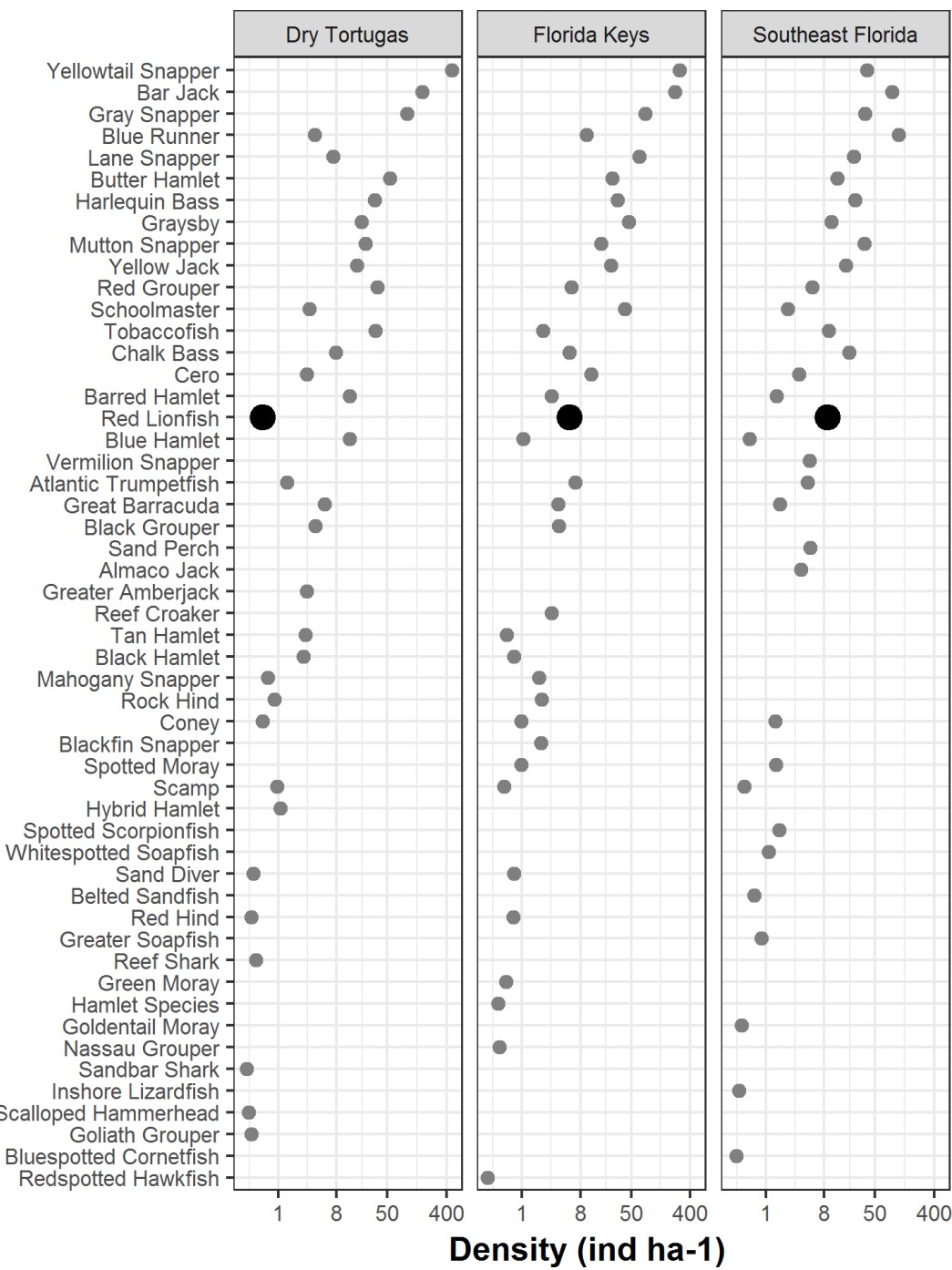

**Figure 4** **Density of piscivorous fish with an occupancy rate greater than 1% in at least one area in 2016.** Lionfish are highlighted by a bold circle and were are ranked 30th out of 66 in the Dry Tortugas, 17th out of 66 in the Florida Keys, and 13th of 68 in southeast Florida. Density (ind ha$^{-1}$) is presented along a logarithmic scale on the $x$-axis.

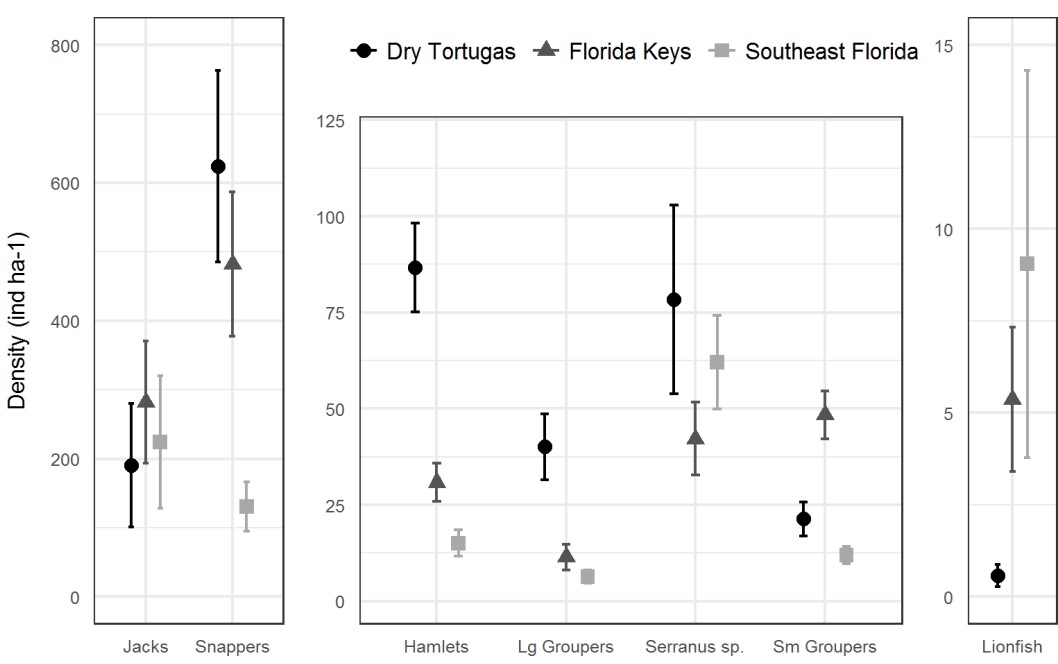

**Figure 5  Regional density estimates (ind ha⁻¹) of lionfish and several groups of potential lionfish predators and competitors in 2016.** Bars represent 95% confidence intervals. *Y*-axis scales are different for each panel.

invasive lionfish are still unclear (*Benkwitt et al., 2017*), and likely depend on the area in question (*Barbour et al., 2011*; *Mumby, Harborne & Brumbaugh, 2011*; *Frazer et al., 2012*; *Hackerott et al., 2013*; *Johnston & Purkis, 2015*). Two principal mechanisms for potential lionfish population control, directed fishing and predation, have been widely discussed in the scientific literature since their introduction into the central western Atlantic and Caribbean Sea (*Barbour et al., 2011*; *Mumby, Harborne & Brumbaugh, 2011*; *Hackerott et al., 2013*; *Smith et al., 2017*). Divers and fishers have become adept at catching lionfish, and while studies have shown some localized benefits, the broader ecosystem impacts of lionfish exploitation are less clear (*Barbour et al., 2011*; *Frazer et al., 2012*; *Green et al., 2014*; *Smith et al., 2017*). The effect of predatory biocontrol to invasive perturbations has been much disputed in the literature, where some areas showed an apparent inverse relationship between predators and lionfish (*Mumby, Harborne & Brumbaugh, 2011*), while others showed no response (*Hackerott et al., 2013*; *Valdivia et al., 2014*). Much of the discussion has been centered on the population effects of large predators on adult lionfish. However, predation or competition at an earlier life stage may have a greater influence on lionfish abundance (*Shulman & Ogden, 1987*; *Hixon, 1991*; *Carr & Hixon, 1995*; *Almany & Webster, 2006*).

The RVC data, designed to provide population-level estimates of reef fish densities, provide a broad, regional perspective on the lionfish invasion in comparison to site or sample-specific estimates commonly found in the literature. Furthermore, differences amongst the south Florida regions can give further insight into the mechanisms controlling

their population. Due to its remote location and numerous large marine protected areas, overall fishing pressure is minimal in the Dry Tortugas, which has resulted in a greater biomass of exploited snapper and grouper species (*Ault et al., 2013*), along with a very low fishing mortality rate of lionfish. This is in contrast to the Florida Keys and southeast Florida, where fishing pressure on native reef fishes is extremely high, snapper and grouper populations are overfished (*Ault, Smith & Bohnsack, 2005*), and divers target lionfish both recreationally and commercially (*Harvey & Mazzotti, 2016*). In addition to greater numbers of large groupers and snappers in the Dry Tortugas, we also found that hamlets and other small serranids (*Serranus* spp.) were more abundant. These small carnivorous fishes have diets that overlap with juvenile and adult lionfish and may be direct competitors during all life stages (*Randall, 1967*; *Whiteman, Côté & Reynolds, 2007*; *Côté et al., 2013*). Some of these potential competitors are piscivorous and may consume lionfish while they are juveniles. Survival of juvenile fish is partly determined by the suite of predators occurring at the site of settlement (*Holmes, & McCormick, 2010*), and early post-settlement mortality through predation is a major driver of reef fish community structure (*Hixon, 1991*; *Carr & Hixon, 1995*; *Almany & Webster, 2006*). Small, active predators have high metabolic and consumption rates which can have a dramatic effect on recently settled juvenile fishes, and hence, a disproportionally large influence on fish assemblages within their range (*Feeney et al., 2012*). From a biological control perspective, it is likely that lionfish populations are more impacted at the early settlement stage from predation than compared to their adult life stage.

In the Florida Keys and southeast Florida, where large piscivores are largely extirpated (*Ault, Smith & Bohnsack, 2005*) and hamlets are rare, fishing mortality may play a larger role in the control of lionfish populations. Although the amount of directed fishing effort and removals of lionfish through the recreational fishery are not well measured, estimated landings in Florida have increased exponentially from 1,040 kg in 2010 to 309,883 kg in 2017 (*National Marine Fisheries Service, 2018a*). Numerous national and local public awareness campaigns have raised awareness on the importance of lionfish removal throughout Florida (*Harvey & Mazzotti, 2016*), presumably leading to greater directed fishing efforts on lionfish. In addition to increased mortality from recreational fishers, the commercial fishery for lionfish in Florida has also grown significantly from 1,080 kg in 2011 to 49,553 kg in 2016 and valued at over $530,000 dollars (*National Marine Fisheries Service, 2018b*). Although fishing mortality has not been high enough to eradicate lionfish (*Barbour et al., 2011*), it appears that in conjunction with potential natural biocontrols, it has been effective in stabilizing lionfish populations in the Florida Keys.

Two other factors that could influence the regional differences in lionfish density are habitat availability and local recruitment processes. Although lionfish have been recorded in a wide range of environments (*Barbour et al., 2010*; *Jud et al., 2011*; *Pimiento et al., 2015*), some research suggests a preference for deeper, higher complexity habitats in invaded reefs (*Biggs & Olden, 2011*; *Claydon, Calosso & Traiger, 2012*). This preference was also observed in south Florida, but the stratification scheme used to optimize the RVC estimates was not conducive for regional comparisons in habitat availability since strata within each region are classified differently and often are comprised of several habitat types. For example, the strata

in the Florida Keys are based on depth, distance from shoreline, and habitat configuration (continuous versus patchy); thus they do not include explicit information on relief or complexity at each site. In the Dry Tortugas, relief and habitat configuration are used to define strata, but depth is not used. In the southeast Florida region, complexity, depth, and habitat configuration is used. Despite these differences in defining characteristics, roughly 25% of each region is comprised of a combination of strata that contain deep and either high relief or high complexity reefs that are preferred by adult lionfish. Given this large amount of available habitat, it is unlikely that lionfish populations are limited by habitat in any region.

Reef fish larval transport and subsequent recruitment in the Dry Tortugas, Florida Keys, and southeast Florida region is largely driven by the Florida Current and its associated eddies (*Lee et al., 1992*; *Sponaugle et al., 2005*). All three regions are highly connected and local retention of larvae plays a major role in recruitment of reef fish (*Lee et al., 1994*; *Sponaugle et al., 2012*; *Bryan et al., 2015*) with limited sources of recruits entering the region from the Caribbean and Gulf of Mexico through the Loop Current (*Roberts, 1997*; *Paris et al., 2005*; *Bryan et al., 2015*). There has been no study looking at differences in recruitment rates between the three RVC regions and previous work on lionfish larval transport has grouped them together (*Johnston & Purkis, 2015*; *Johnston, Bernard & Shivji, 2017*). Since regional rates of lionfish recruitment are unknown, it cannot be ruled out as a possible factor influencing the differences in density.

Although lionfish populations in south Florida appear to have stabilized, their successful colonization could still have major direct and indirect impacts on the local fish community (*Black et al., 2014*; *Benkwitt, 2015*; *Kindinger & Albins, 2017*; *Sancho et al., 2018*). Their density was in the top 10% for piscivores in the southeast Florida region, and in the top 25% in the Dry Tortugas and the Florida Keys. Currently it is unknown if their establishment in south Florida has caused a regional shift in the fish community. In other invaded areas off the southeastern coast of the United States, where lionfish densities are greater, there has been a regional reduction in the native tomtate population (*Ballew et al., 2016*), whereas on the Belize Barrier Reef no effect on prey species was observed (*Hackerott et al., 2017*). Additional analysis utilizing current and future RVC data may be useful to determine if lionfish have had any effect on local reef fish communities in south Florida.

It is also important to note that while the RVC survey domain covers a large extent of lionfish habitat in south Florida, it is possible that trends in lionfish density are different in waters deeper than 35 m, or in shallow seagrass or mangrove habitats. Lionfish have been commonly found on mesophotic reefs (40–150 m) in the Gulf of Mexico and Bahamas (*Lesser & Slattery, 2011*; *Nuttall, 2014*), and are anecdotally common on deep reefs and artificial structures throughout south Florida, but little is known of their density or possible trends in abundance. Standardized surveys of mesophotic and artificial reefs would provide valuable information to further understand the population status of lionfish in south Florida.

## CONCLUSION

Lionfish have become an established predator in the south Florida reef fish community. The population density has been relatively stable along the south Florida coral reef tract in waters less than 35 m depth. Relatively low and stable density estimates suggests a mechanism for population control in south Florida. In the Dry Tortugas, where lionfish densities are lowest, fishing pressure is minimal but predators and competitors are more abundant, suggesting that lionfish population density may be naturally controlled. In the Florida Keys and southeast Florida, fishing pressure is extremely high, and the resultant density of predators and competitors is only slightly lower, implying that a combination of fishing mortality and biocontrols may be constraining lionfish population growth.

## ACKNOWLEDGEMENTS

The RVC reef fish visual census is a long term and large scale monitoring effort that has only been possible through extensive partnerships and collaboration involving a number of federal and state agencies and institutions: NOAA's Southeast Fisheries Science Center, University of Miami, National Park Service, Florida Fish and Wildlife Conservation Commission, and Nova Southeastern University. We greatly appreciate the contributions of literally hundreds of scientific divers that have participated in the program since its inception. We thanks Steven G. Smith for statistical support and Harry Ganz for creating the R open source code RVC package used here for analyses.

### Funding

David R. Bryan and Jerald S. Ault received funding from the National Park Service, Grant # P16AZ01569, and the National Oceanic and Atmospheric Administration, Cooperative Institute for Marine and Atmospheric Studies, Grant # NA15OAR4320064. The funders had no role in study design, data collection and analysis, decision to publish, or preparation of the manuscript.

### Grant Disclosures

The following grant information was disclosed by the authors:
National Park Service: #P16AZ01569.
National Oceanic and Atmospheric Administration, Cooperative Institute for Marine and Atmospheric Studies:  # NA15OAR4320064.

### Competing Interests

The authors declare there are no competing interests.

### Author Contributions

- David R. Bryan conceived and designed the experiments, performed the experiments, analyzed the data, contributed reagents/materials/analysis tools, prepared figures and/or tables, authored or reviewed drafts of the paper, approved the final draft.

- Jeremiah Blondeau performed the experiments, analyzed the data, contributed reagents/materials/analysis tools, prepared figures and/or tables, authored or reviewed drafts of the paper, approved the final draft.
- Ashley Siana conceived and designed the experiments, prepared figures and/or tables, authored or reviewed drafts of the paper, approved the final draft.
- Jerald S. Ault conceived and designed the experiments, authored or reviewed drafts of the paper, approved the final draft.

## Data Availability

All data used for this manuscript and accompanying code are available in a Supplemental File.

## Supplemental Information

Supplemental information for this article can be found online at http://dx.doi.org/10.7717/peerj.5700#supplemental-information.

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
