# Peer review of "Regional differences in an established population of invasive Indo-Pacific lionfish (Pterois volitans and P. miles) in south Florida"

_PeerJ, doi:10.7717/peerj.5700_

## Round 0.1 · original submission · Major Revisions

I now have detailed comments back from two expert referees who are equally appreciative of the extensive data set compiled in this submission.

However, while supportive of the work, each also has similar concerns about the manuscript in terms of comparisons among different areas in terms of sampling methodology and habitat type. Each referee raised a number of concerns about specific analyses, data comparisons, or scholarship that to me constitute a major revision. There are differing suggestions from each referee about the focus of the paper, and I am inclined to leave this to the authors so long as you can address the criticisms about whether the abundance estimates in Florida are at all comparable to those from other locations that you have included here. I tend to agree with the referees that differences in methodology and habitat type could have a significant impact that must be addressed to avoid similar reactions from future readers of the manuscript.

Reviewer 1 ·

Basic reporting

This is a well written manuscript that adds depth to our understanding of the size of the population of lionfish in the invaded western Atlantic region. One of the main strengths of this manuscript is the scope, in terms of area surveyed, which is substantially larger than other studies of lionfish population sizes in the Atlantic that I am familiar with. However, the authors over emphasize the potential for depredation and harvesting explaining the lower densities of lionfish than expected without accounting for other variables that could be influencing the overall pattern they observe.

Experimental design

The survey design is appropriate for investigating the density of lionfish found in the regions survey and allows for the authors to compare these densities to the density of native fishes. Since the surveys go back to 1979 (Line 80) the authors could compare densities of native species before and after the lionfish invasion to see if there are differences in the native community coincident with the invasion of the lionfish. While an analysis of all fishes may be beyond the scope of this manuscript, specifically looking at changes in the competitor/predator densities could support the discussion of the role of competitors/predators in regulating the densities of lionfish.

Validity of the findings

The authors primarily focus on the likely influence of competitors/predators coupled with human exploitation as the primary explanation for the observed density of lionfish being lower than expected. However, there are other possible reasons that should be addressed either to show that the expectation from those explanations differs from what is observed or to show that they may be other explanations which need further investigation. For example, habitat may play a role in the patterns observed. The authors state that there are differences in habitat types among all three surveyed locations as well as within locations (Lines 98-118). From my understanding, these differences in habitat are pooled together to determine regional densities of lionfish (Lines 142-144). However, when comparing among regions and to other literature there doesn’t appear to be an analysis of differences in densities observed based on habitat type. It may be for example that the densities of lionfish in their “preferred” habitat in the survey area is similar to elsewhere in the Caribbean and the literature reviewed consistently surveys more preferable habitat without incorporating the less desirable habitat thus accounting for the differences. An analysis of the lionfish density as it relates to the surveyed habitat type would be ideal to see if A) there are differences among habitats which may explain some of the pattern observed within this survey and B) help in improving the ability to compare this study with other literature.

In lines 295-302 the authors posit that there may still be an effect of lionfish on native fish densities despite the relatively low densities of lionfish. The authors should consider utilizing the model developed by Green et al. (2014) in order to probabilistically determine if there is likely to be an effect of lionfish given the current health of the native fish populations and current lionfish densities. This would be helpful to guide future research and would provide an interesting test case for the model Green et al. develop. Further, if the authors choose to perform the analysis I mentioned above this model could be incorporated into that analysis to see the potential for local effects of lionfish beyond simply the regional scale, though this may well be beyond the scope of this manuscript.
Green SJ., Dulvy NK., Brooks ALM., Akins JL., Cooper AB., Miller S., Côté IM. 2014. Linking removal targets to the ecological effects of invaders: a predictive model and field test. Ecological Applications 24:1311–1322. DOI: 10.1890/13-0979.1.

Additional comments

This is an excellent manuscript documenting the invasion of lionfish into south Florida. The discussion focuses primarily on the potential role of humans and competitors/predators on maintenance of the low densities of lionfish observed. This should be explored further to support the supposition that these are the prime factor in controlling lionfish density or other factors should also be explored as depredation/exploitation is only one of a myriad of potential causes for the lower densities, habitat being the most obvious example I can think of.

Reviewer 2 ·

Basic reporting

See below

Experimental design

See below

Validity of the findings

See below

Additional comments

Review of Bryant et al

In this paper, the authors examine 9 years of lionfish data from the Florida Reef Tract, beginning in 2009 when the invasion was first detected in the Florida Keys. The authors use an existing monitoring dataset to examine differences across regions. Interestingly, they note that the Dry Tortugas, which is more remote and more protected from fishing, and therefore has higher abundances of predatory fish, has lower abundance of lionfish. This encouraging result suggests that native predators may be able to at least partly control the lionfish invasion. While other work has suggested this may be true, this is the most compelling dataset to date since it covers a large area. I believe this manuscript will make a useful contribution to our understanding of the lionfish invasion.

That said, there are a few issues that require attention.

First, the sampling methodology employed in this study likely underestimates the actual abundance of lionfish, since lionfish can be cryptic and this method targets conspicuous reef fishes. The methods employed in this study are not lionfish specific, and therefore likely underestimate the actual abundance, particularly relative to lionfish-specific surveys used elsewhere. Therefore, the comparisons between Florida and other locations may not be valid, and the burden of proof is on the authors to demonstrate that their data are comparable. For example, REEF has conducted lionfish-specific surveys in the area; how well do those estimates match the ones presented here?

That said, data describing relative differences within part of the Florida Reef Tract where these methods are used consistently is useful, and I strongly encourage the authors to focus their framing and discussion on comparisons within their dataset. It is ok to mention other locations, I think this should occur only in the discussion and should be concurrent with a discussion of these methodological issues, emphasizing the uncertainty of such comparisons to elsewhere in the Caribbean or Indo-Pacific.

Second, the authors should consider other factors that may impact lionfish abundance, such as habitat and area. For example, the Dry Tortugas is fairly isolated oceanographically, such that there may be significant self-recruitment. Low abundance of lionfish there may provide fewer recruits than elsewhere. The Florida Keys may receive recruitment from the Keys themselves and from the Dry Tortugas. In addition, Lionfish were first detected in the SE FL region in 1985, so they have been established there much longer than FK or DT, and may also receive recruitment from downstream locations in the Keys and Dry Tortugas.

Habitat may also impact lionfish abundance; the complete patch-reef forereef structure in the Keys may serve as better lionfish habitat than many areas in the Dry Tortugas, so some examination of habitat-specific densities would be helpful.

While the potential predator control effect is compelling, the authors need to acknowledge additional factors that may drive the patterns they observe in these data from a large-scale survey program.

The scholarship also needs improvement. There are many places where citation of related work is needed; in some cases, citations need to be broader than they are, and other cases, entire statements require a citation or a few. I suggest a critical read through the MS, and that the authors include additional references where appropriate.

Specific comments:

L75: I like this approach; THIS should be the focus of the paper, not the comparisons to other Caribbean/Indo-Pacific locations
L126-152: These methods are described in detail in Smith et al 2011; these methods could be significantly condensed. At the very least, the formulae and detailed descriptions could go in supplementary materials.
L179: Citation needed (an example; there are several other locations where the scholarship could be strengthened
L190-192: Again, I am not convinced that these comparisons are appropriate, and despite this caveat, the authors make these comparisons explicit in Fig 3 (which I believe should be removed)
L201-202: The sharp decline suggests sampling error to me, and possibly detection error by new divers since lionfish are difficult to detect. How much were observers trained to search for lionfish from year to year? If it is real, this decline is very intriguing, but again, the authors need to convince me that it is real.
L213-215: Once again, I am not convinced comparisons to other regions are valid, and in any case, there are no statistics to support a claim that densities are ‘lower.’ The authors need to convince me that the comparisons are valid in the first place and provide some way of evaluating differences. The simplest solution would be to remove these comparisons
L219-221: Again, how different were these survey methods? Were methods in Kulbicki 2012 lionfish specific? If so, these densities are almost certainly not comparable
L238-240: Again, I'm not sure I believe these comparisons, since methods differ.
L240-243: I don't think the data support this interpretation at all.
L247-249: Citations needed.

Figures:
Fig 3: I would remove figure 3 for the reasons described above.

Figs 4 and 5: Why not use biomass for predatory species? Biomass integrates size and abundance, and might be a better proxy for potential lionfish predators than numerical density.

---

## Round 0.2 · Minor Revisions

Thank you for your revisions. Overall your manuscript is now acceptable, although the referee points out a number of potential typos (particularly in regards to the equations) that would hinder understanding by readers. I expect that you would prefer to correct any mistakes or clarify the text to prevent future misunderstandings, but leave it to you to decide which of the other comments to incorporate (or not) in your final manuscript before it moves into production.

Reviewer 1 ·

Basic reporting

In this manuscript, the authors present important data regarding the population density of lionfish in various regions of Florida. The authors have done a good job addressing my previous comments and concerns. Below are some fairly minor changes I feel would improve the manuscript and one addition needed in the methods.

Experimental design

No Comment

Validity of the findings

L204-216: You don't mention in the methods how the differences between strata within regions were statistically tested. If you used the same method as you did for statistically analyzing differences between regions (95%CI t-test) then that should be mentioned in the methods. If not then the method that was used should be mentioned.

Additional comments

Minor Comments:

L142: In this equation (and the equation in L151) I believe the subscript 'st' is to denote that the density is for the region. However, previously different subscripts had been used that did not include a region subscript. I would find it clearer if instead of introducing new and undefined subscripts you simply rephrase L132 to say something like "the mean density for each PSU (i) in stratum (h) in region (r)" and then alter the formulas to represent this as the labeling convention to maintain consistency across all the various equations.

L142: Are there meant to be 3 horizontal lines over the "Dst" in this equation since it is the third mean (SSU to PSU then PSU to Strata then Strata to Region)?

L145 &147: The equations on these two lines appear to be identical and from the context of the methods I believe they are meant to be different (specifically the subscripts).

L217-220: It would be helpful to report the total number of species as well as the ranking of lionfish. For all I know reading this section there are a total of 134 species in the Dry Tortugas, 95 species in the Florida Keys and 78 species in southeast Florida, this would be interesting of course but would be a far different message than what I believe this paragraph is coveying (i.e. that lionfish are fairly common but not the dominant species numerically). The same holds for the number of piscivores present.

L232: Change "have appeared" to "appear"

L233-236: Because you removed the figure comparing lionfish densities in this study to previous studies it may be a good idea to include some of the values presented for the native and invaded range in the text since you do mention the comparison as something of interest here.

L238: Change "depends" to "depend"

L264-265: Is the diet overlap between small piscivorous fishes and lionfish similar at all life stages (of the lionfish) or at particular life stages? If it is specifically similar to the diet of juvenile lionfish then that would be worth mentioning especially given the apparent transition from crustaceans to fishes as lionfish age (Morris and Akins 2009, Dahl and Patterson 2014).

Figure 3 Legend: One of the abbreviations is "CONT_MR" that is said to stand for "continuous low relief". In later abbreviations, low relief strata are termed "_LR". Is this meant to be abbreviated as "CONT_LR" or is it a "continuous medium relief site"?

Figure 3: This figure does not aesthetically match the other figures (e.g. background colour). I would suggest changing this so all the figures are similar aesthetically.

---

## Round 0.3 · accepted · Accept

Thanks for clearing up those final editorial issues and clarifications of the manuscript. I am happy to accept your paper and move it forward into production. Thanks for selecting PeerJ to publish your work.

#